# Perceptions, Knowledge and Adaptation about Climate Change: A Study on Farmers of Haor Areas after a Flash Flood in Bangladesh

**Kanis Fatama Ferdushi** [1,*] , **Mohd. Tahir Ismail** [2] **and Anton Abdulbasah Kamil** [3]

1   Department of Statistics, Shahjalal University of Science and Technology, Sylhet 3114, Bangladesh
2   School of Mathematical Sciences, Universiti Sains Malaysia, USM Pulau Pinang 11800, Malaysia
3   School of Industrial and System Engineering, Telkom University, Bandung 40257, Indonesia
*   Correspondence: kanisusm@gmail.com

**Abstract:** Bangladesh remains one of the most vulnerable countries in the world to the effects of climate change. Given the reliance of a large segment of the population on the agricultural sector for both their livelihoods as well as national food security, climate change adaptation in the agricultural sector is crucial for continued national food security and economic growth. Using household data from lowland rice farmers of selected haor areas in Sylhet, the current work presents an analysis of the determinants behind the implementation of different climate change adaptation strategies by lowland rice farmers. The first objective of this study was to explore the extent of awareness of climate change within this population as well as the type of opinions held by lowland rice farmers with respect to climate change. To serve this purpose, a severity index (SI) was developed and subsequently employed to evaluate the perceptions and attitudes of 378 farmers with respect to climate change vulnerability. Respondents were interviewed with respect to climate change related circumstances they faced in their daily lives. Attained SI index values ranged from 69.18% to 93.52%. The SI for the perception "Climate change affects rice production" was measured as 93.52%. Using data collected from the same 378 farmers, a logistic regression was carried out to investigate the impact of socio-economic and institutional factors on adaptation. The results show that credit from non-government organizations is highly statistically significant for adaptation, and that rural market structure also has a positive effect on adaptation. Among the studied factors, credit from non-governmental organizations (NGOs) was found to be the most important factor for adaptation. The results of this work further indicate that marginal farmers would benefit from government (GoB) funded seasonal training activities that cover pertinent information regarding adaptation after flash floods. Additionally, the authors of this piece recommend timely issuance of government-assisted credit during early flash floods to afflicted farmers, as such an initiative can aid farmers in adapting different strategies to mitigate losses and enhance their productivity as well as livelihood.

**Keywords:** perception; severity index; adaptation planning; rice farmers; flash flood

## 1. Introduction

Owing to its geographical location, climate, and topology, Bangladesh is characterized by agro-zones that are highly susceptible to drought, cyclones, flooding, and rising salinity, rendering Bangladesh one of the most vulnerable countries in the world to climate change. Such findings have been extensively supported in the literature [1,2], particularly in relation to sea-level rise and intense climatic events. Indeed, climate change poses extensive economic and physical risk to all societies by negatively impacting a number of factors, such as water resources, agriculture, and terrestrial

and marine ecosystems, among others. However, in developing nations such as Bangladesh, where agriculture plays a central role in the economy by providing subsistence and income to a large segment of society, the effects of climate change pose an even more serious economic risk, particularly for nations consisted of agro-zones susceptible to submersion.

In an attempt to mitigate the effects of climate change on agriculture and the environment, farmers worldwide, from both technology-advanced and emerging economies, have begun to adopt new ideas and technologies within their agricultural practice in tandem with traditional farming techniques. Of these, a large number of farmers from different agro-zones have begun to adapt strategies based on native adaptation evidence and capacities, namely, their local and indigenous knowledge. According to United Nations Educational, Scientific and Cultural Organization's (UNESCO) programme on Local and Indigenous Knowledge Systems, local and indigenous knowledge refers to the understandings, skills, and philosophies developed by societies.

Increased agricultural productivity through adoption and diffusion of modern agricultural technologies is one of the key pathways to monetary growth and transformation in emerging countries [3–5]. Within this context, previous studies have attempted to investigate agricultural adaptation strategies related to climate change in Bangladesh, a difficult task given that the agricultural system of Bangladesh is composed of geographically-diverse zones vulnerable to a myriad of circumstances, such as submersion, drought, increased salinity due to sea intrusion, river erosion, and embankment damage, among others. Bearing in mind the complexity of the agricultural system in Bangladesh, [6] has emphasized the importance of implementing native adaptation strategies aimed at different climatic zones, e.g., submersion-prone, drought-prone, and salinity-prone zones of Bangladesh. According to [7] integrating adaptation into development is indispensable for sustainable development to occur in developing nations. Thus, in order for Bangladesh to become and remain self-sufficient and food-secure, especially after natural disasters such as flash floods, sufficient climate change adaption measures must be implemented within the agricultural sector. Among these, the dissemination of adaptation knowledge and wisdom plays a key role in strengthening the resilience of Bangladesh's agricultural sector, as such a practice will aid in safeguarding farmers' livelihoods and agricultural sustainability. Implementation of adaptation practices can yield positive results for farms, enabling economic growth of individual agricultural farmers, as well as the country as a whole. In this regard, smallholder farmers currently experiencing the effects of climate change have begun to adapt diverse strategies through their native agricultural practices [8,9]. Adaptation planning is interconnected to farmers' knowledge, awareness, and perceptions about climate change. As such, a clear understanding of climate change is necessary for farmers to implement appropriate adaptation strategies and techniques by [10]. Here, the theme 'perceptions of climate' refers to the ways in which climate is regarded, understood, or interpreted by farmers. It can be defined as farmers' awareness about climate change through their senses. Farmer's perceptions regarding climate influence how farmers choose to adapt to the impacts of today's changing climate on their livelihoods. Farmers that perceive climate change as an occurring phenomenon that is human-induced in nature are more likely to seek positive adaptation strategies. Indeed, previous studies have indicated that most farmers perceive the climate as changing, and correspondingly adapt to reduce the negative impacts of climate change [11–13]. Studies further show that awareness of climate change [14–16] and the adoption of adaptive measures [17,18] are influenced by different socio-economic factors. As well, according to [19], factors such as personal experience, local knowledge, familiarity, and social-learning exchanges between farmers and the public may help to boost mutual understanding and to reduce agricultural systems vulnerability. In the current work, perceptions of farmers towards climate change were converted to a severity index (SI), which is a scaling tool to measure the relative strength of a behavior, attitude, or occurrence. This type of indexing has not been used before in the context of Bangladeshi agriculture.

Sustainable development within the Bangladeshi agricultural sector hinges upon farmer's beliefs, attitudes, and knowledge regarding climate change, and their resulting present and future planning in view of our changing climate. Sustainability, peaceful agriculture, rural development, and social

protection are important measures for the overall development of Bangladesh. Social inclusiveness and information-sharing aim at empowering farmers through the dissemination and sharing of knowledge so that they can take initiative and make decisions to improve their agricultural practice. Climate change adaptation (CCA) in the lowland areas of emerging countries has become essential for continued maintenance of food availability, production, and sustainability. The haor areas of sylhet are susceptible to submersion, and their inhabitants have been facing adverse conditions related to flooding every year within recent years, demanding that measures are taken to better understand and address this issue. In the sampling year 2017, the Department of Agriculture Extension (DAE) reported that food availability was greatly affected by low production and productivity due to factors linked to climate change in Bangladesh. Losses in agricultural production stemming from damage due to flooding renders farmers helpless while also creating food insecurity within the country for periods of time, as considerable amounts of national rice production, as well as many livelihoods, depend on agricultural production stemming from these haor areas. In these moments of crisis, farmers have utilized locally acquired knowledge to engage in small scale adaptation practices and production diversification as strategies to mitigate the impact of existing climate conditions. The currently presented study surveyed data from the most severely inflated embankment and flood-prone district in Bangladesh with the aim of attaining information on local perceptions of climate change, which principally affects farmers' adaptation approaches.

Using this framework, we tried to accomplish the following objectives:

(1) To explore the extent of awareness regarding climate change, including farmers' perceptions and attitudes about climate change, as such factors are significantly associated with positive adaptation of lowland rice farmers to climate change and variability;
(2) To identify farmers' knowledge about climate change vulnerability;
(3) To determine, among others, the socioeconomic factors that significantly affect farm-level adaptation strategies due to climate change in the context of lowland rice farmers;
(4) To analyze farm-level adaptation strategies after early flash floods due to changing climate conditions.

## 2. Materials

### 2.1. Study Areas

Six districts (Sylhet, Moulauvibazar, Sunamganj, Habiganj, Netrokona, and Kishoreganj) were affected in the northeast region of Bangladesh by early flooding that began on 28 March 2017. Rising water overflowed and breached embankments, inundating vast areas of croplands and damaging approximately 160,170 hectares (hec) of nearly ready-for-harvest boro rice, putting the livelihoods of thousands of farmers at risk. According to [20], the Sunamganj district was the most affected among the abovementioned six districts, with 11 upazilas (upazilas are the second lowest tier of regional administration in Bangladesh, and function as sub-units of districts) affected by early flash floods. Sunamganj, which has come into international focus for various reasons, is a unique wetland ecosystem composed of many haors (bowl or saucer shaped depressions that resemble back swamps) and beels (lake-like wetlands; generally smaller than haors), and of significant national importance. The Sunamganj district has an annual rainfall of 3334 mm, and reaches a maximum annual average temperature of 33.2 °C and a minimum 13.6 °C. Among the 10 upazilas in Sunamganj, Dharmapasha, located in the Tanguar haor, was most severely affected by the abovementioned early flooding. The total boro rice crop damage was estimated at 18,610 hectares out of a total of 31,800 planted hectares. Embankments in Ajarkhali, Ulashkhali, Balrampur, and Haldirbodh, located in the Dharmapasha upazila, were among the most affected areas, and thus herein considered as regions in need of significant adaptation strategies. Study zone maps are given in Figure 1.

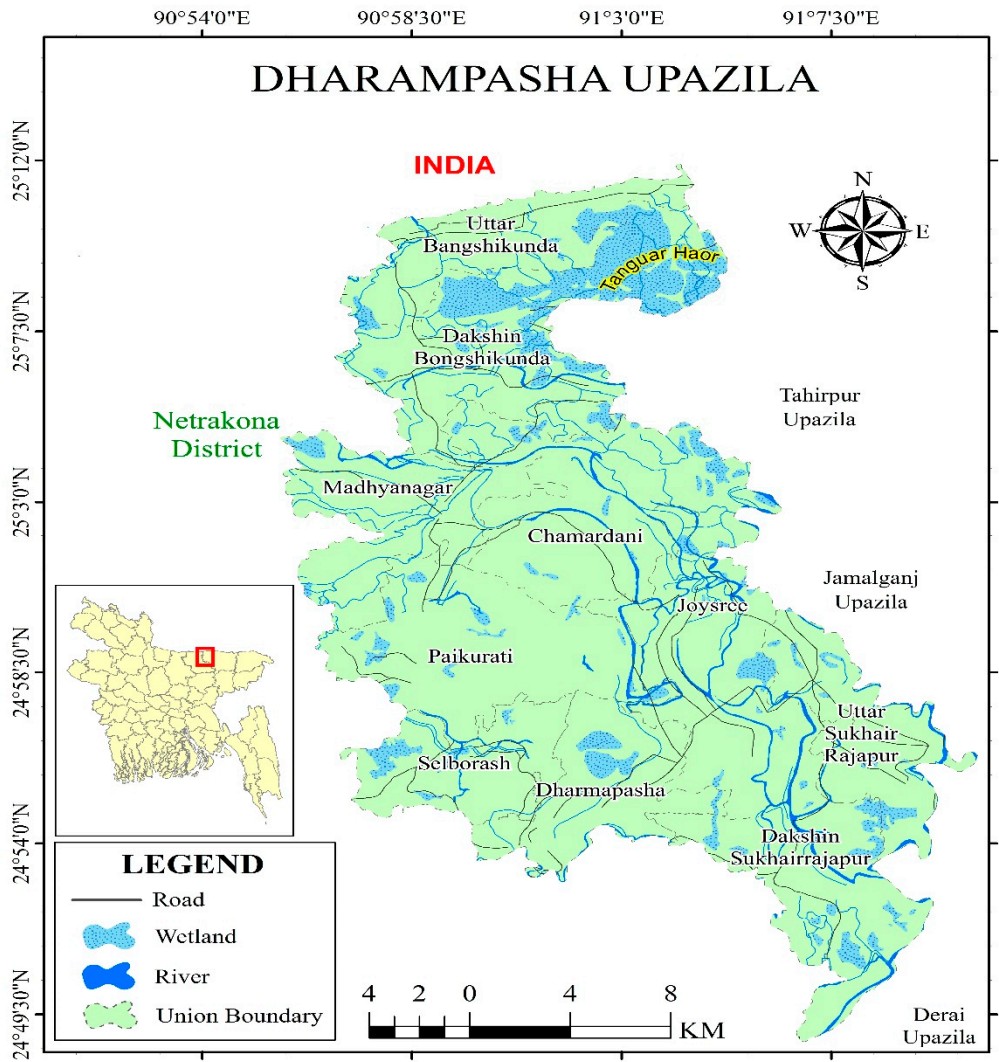

**Figure 1.** Map of Dharmapasha Upazila. Study areas are marked in the Upazila map.

*2.2. Sample Size*

The current study employed a cluster sampling technique in which census enumeration areas comprising 100–120 households, as defined by the Bangladesh Bureau of Statistics (BBS), were considered as a cluster. These census enumeration areas have been previously considered as clusters in a number of past studies, including [21]. The current study covered 4 clusters purposely selected to fulfil the objectives of the study. Three randomly selected villages were considered from four embankments or clusters. The recognized sample size determination formula for the villages was $n = z^2 \left[ p(1-p)/d^2 \right] * D_{eff}$; where $p$ is the indicator percentage, $Z$ is the value of normal variate with a 95% confidence interval, and $D_{eff}$ is the design effect. Attained values were calculated on the basis of 50% indicator percentage (proportion of households adapting new technologies), 95% confidence interval, 0.13p relative precision, and highest response distribution with an assumed design effect of 1.85. Using this design, a total of 105 households were calculated as the substantial number sample for each cluster. The sample size determination formula yielded at least 420 data form clusters. Due to migration in villages considered within the cluster sample area, data from only 378 farmers were collected in interviews. In total, 129, 103, 81, and 65 farmers belonging to the villages of Ajharkhali, Balrampur, Haldirbodth, and Ulashkhali, respectively, were interviewed.

## 2.3. Sampling Technique and Data Collection

A structured interview schedule containing both close-ended and open-ended questionnaires was developed and subsequently employed on Nov, 2017 to collect pertinent data from 378 farmers. In addition to the above mentioned interview data, the study also utilized primary cross-sectional data. Quantitative techniques were employed to analyze the data.

## 3. Methodology

### 3.1. Severity Index

In [22] introduced the severity index, which can be calculated using the following formula:

$$SI = \sum_{i=0}^{4} p_i q_i / n \sum_{i=0}^{4} q_i$$

where $p_0$, $p_1$, $p_2$, $p_3$, $p_4$ represent the response frequencies (perceptions and attitudes) corresponding to a 5-point Likert Scale $q_0 = 0$, $q_1 = 1$, $q_2 = 2$, $q_3 = 3$, $q_4 = 4$. Here, $n$ = total number of respondents against a 5-point Likert Scale.

In this work, [22] severity index evaluation arrangement was used to measure farmers' perceptions regarding climate change. Collected data was analyzed using the following criterion:

$q_0$ = Strongly disagree, $0.00 \leq SI < 12.5$;
$q_1$ = Disagree, $12.5 \leq SI < 37.5$;
$q_2$ = Moderate, $37.5 \leq SI < 62.5$;
$q_3$ = Agree, $62.5 \leq SI < 87.5$ and;
$q_4$ = Strongly agree, $87.5 \leq SI < 100$.

In order to investigate two distinct yet interrelated parameters, namely (a) farmer's perceptions and attitudes regarding climate change, and (b) their overall knowledge regarding climate change vulnerability, related items corresponding to either one of these categories in the questionnaire were clustered into one of the following two groups:

Group 1: Perceptions and attitudes regarding climate change issues.
Group 2: Knowledge of climate change vulnerability.

### 3.2. Chi-Square Statistic

The formula for the chi-square statistic used in the chi-square test is as follows:

$$\chi_c^2 = \sum \frac{(O_i - E_i)^2}{E_i}$$

Here, $c$ is the degree of freedom, $O$ is the observed count/frequency (row and column frequencies), and $E$ is the expected value, where $E = \frac{(Row\ total)\ (Column\ total)}{Total\ of\ Both\ Samples}$, while $\sum$ indicates the sum of the square variances between $O$ and $E$ for all data items. The chi-squared statistic is a single number that is used to determine whether there is a significant difference between observed counts and expected values. A *p*-value based on the chi-squared number and the degrees of freedom will denote whether the attained results are statistically significant.

### 3.3. Bivariate Logistic Regression

Bivariate logistic regression was used to investigate the effect of socio-economic factors on adaptation. Here, categorical responses have two possible outcomes, $Y$; 1 = having adapted, and 0 = otherwise, and a group of predictor variables, where $p$ is the probability of $Y$ being 1, $P = P(Y = 1)$.

Let $x_1, x_2, \ldots, x_k$ be a set of predictor variables. The logistic regression of $Y$ on $x_1, x_2, \ldots, x_k$ estimates parameter values for $\beta_0, \beta_1, \ldots, \beta_k$ through the maximum likelihood method, which uses the following equation:

$$\log it(P) = \log\left(\frac{P}{1-P}\right) = \beta_0 + \beta_1 x_1 + \ldots + \beta_k x_k$$

where $P/1-P$ denotes the odds ratio in favor of having adapted strategy; in other words, the ratio of the probability that a farmer will have successfully adapted to the probability that a farmer will not have adapted after a flood.

### 3.3.1. Dependent Variable

Adaptation is considered as the dependent variable of this study. Adaptation management processes depend on many factors, including who or what adapts, what they adapt to, how they adapt, as well as what and how resources are used, among many other themes [23]. Adaptation is understood in this study to encompass all risk-management strategies undertaken in response to adverse conditions; in the case of this study, in response to flash floods. After the flash flood, farmers were asked about their opted adaptation strategies. Have they adopted new strategies? Or have they been unable to mitigate their situation? Variable adaptation was coded as 1 if they had adopted adaptation strategies, or as 0, if otherwise.

### 3.3.2. Predictor Variables

The predictor variables included in this study encompassed socio-economic variables, farm information variables, and institutional and market accessibility variables. Gender, age, education, family size, income, land ownership, and farming experience were considered as socio-economic variables, while off-farm earning, livestock ownership, cultivation of flood-tolerant rice varieties, short-duration boro rice cultivation, and cultivation of non-rice rabi crops were considered as farm information variables. Finally, distance to local market, subsidies for seed after flood, credit from NGO/GoB, and rural market structure were considered as institutional and market accessibility variables. All of the above-mentioned variables were considered as possible predictor variables and included in this study.

### 3.4. Weighted Average Index

A weighted average (WA) is a type of average where each observation in the data set is multiplied by an assigned weight reflecting its importance prior to summing all data into a single average value.

$$WAI = \sum w_i x_i / \sum w_i$$

where $w_i$ indicates respective weights for the items.

For the four different categories under consideration in this study, the equation is as follows:

$$WAI = (LM * 0 + LI * 1 + SL * 2 + LL * 3)/N$$

where *LM* = Limited money, *LI* = Lack of information, *SL* = Shortage of labor, and *LL* = Lack of land.

This WA index was herein used to identify farmers' ranking of adaptation practices.

## 4. Results

### 4.1. Farmers' Perceptions and Attitudes towards Climate Change

Respondents were asked about circumstances they faced in their daily lives with respect to climate change. Findings are shown in Table 1. Severity Index (SI) values ranged from 69.18% to 93.52%. An SI value of 93.52% was attained for the perception "Climate change affects rice production" (SI = 93.52%).

The SI value for the perception "New disease appears in agricultural crops" was calculated at 69.18%, while the perception "Climate change already affects the Bangladeshi agricultural sector" yielded a value of 66.49%, indicating that for the most part, farmers agreed with these two statements. The SI value for "Climate change is already affecting my local climate" was calculated at 84.85%. For the perceptions "Precipitation is increasing" and "Cyclones are increasing", SI values corresponding to 86.44% and 89.55%, respectively, were attained. An SI value of 71.10% was attained for the attitude "I feel adaptation has become necessary for all of us". The above findings thus indicate that generally speaking, most farmers perceive agriculture to be affected by climate change and that climate change adaptation measures are necessary for successful agricultural practice and sustainable livelihood.

**Table 1.** Farmers' perceptions and attitudes with respect to climate change issues.

| Descriptions of the Selected Items | | SD (0) | DA (1) | MA (2) | A (3) | SA (4) | SI (%) |
|---|---|---|---|---|---|---|---|
| Climate change affects rice production | NRS | 1 | 1 | 3 | 85 | 288 | 93.52 |
| | PRS | 0.3 | 0.3 | 0.8 | 22.5 | 76.2 | |
| New diseases appear in agricultural crops | NRS | 4 | 82 | 25 | 154 | 113 | 69.18 |
| | PRS | 1.1 | 21.7 | 6.6 | 40.7 | 29.9 | |
| Drought is increasing | NRS | 222 | 156 | 0 | 0 | 0 | 41.27 |
| | PRS | 58.7 | 41.3 | 0 | 0 | 0 | |
| Climate change is a serious problem | NRS | 0 | 0 | 29 | 177 | 172 | 84.46 |
| | PRS | 0 | 0 | 7.7 | 46.8 | 45.5 | |
| Climate change already affects the Bangladeshi agricultural sector | NRS | 0 | 46 | 105 | 159 | 68 | 66.47 |
| | PRS | 0 | 12.2 | 27.8 | 42.1 | 18.0 | |
| Climate change is already affecting my local climate | NRS | 0 | 1 | 29 | 168 | 180 | 84.85 |
| | PRS | 0 | 0.3 | 7.7 | 44.4 | 47.6 | |
| Climate change will have a direct impact on me | NRS | 0 | 0 | 27 | 161 | 190 | 85.78 |
| | PRS | 0 | 0 | 7.1 | 42.6 | 50.3 | |
| Precipitation is increasing | NRS | 0 | 0 | 26 | 153 | 199 | 86.44 |
| | PRS | 0 | 0 | 6.9 | 40.5 | 52.6 | |
| Cyclones are increasing | NRS | 1 | 3 | 20 | 105 | 249 | 89.55 |
| | PRS | 0.3 | 0.8 | 5.3 | 27.8 | 65.9 | |
| I feel adaptation has become necessary for all of us | NRS | 2 | 11 | 110 | 176 | 79 | 71.10 |
| | PRS | 0.5 | 2.9 | 29.1 | 46.6 | 20.9 | |

Note: NRS, PRS, SD, DS, MD, A, SA indicate the number of respondents, percentage of respondents, strongly disagree, disagree, moderate, agree, and strongly agree, respectively, for *n* = 378 farmers.

## 4.2. Farmers' Knowledge about Climate Change Vulnerability

Respondents were surveyed on their knowledge about climate change and vulnerability. The attained results are given in Table 2. Severity indexes corresponding to 74.93%, 62.50%, 50.40% and 38.49% were respectively attained for the surveyed items "Climate is changing", "Consequences of climate change", "Major climate events" and "Reasons behind climate change". Among the respondents, 170 farmers had no knowledge of the causes behind the current increases in drought and flood within the region, while 174 respondents had no comment about this issue, which would indicate they had insufficient knowledge about climate change at the time of the survey.

**Table 2.** Farmers' knowledge about climate change vulnerability.

| Descriptions of the Selected Items | | NC (0) | DK (1) | NM (2) | K (3) | WK (4) | SI (%) |
|---|---|---|---|---|---|---|---|
| Climate is changing | NRS | 1 | 3 | 12 | 342 | 20 | 74.93 |
| | PRS | 0.3 | 0.8 | 3.2 | 90.5 | 5.3 | |
| Consequences of climate change | NRS | 0 | 5 | 181 | 190 | 2 | 62.50 |
| | PRS | 0 | 1.3 | 47.9 | 50.3 | 0.5 | |
| Causes of climate changing | NRS | 0 | 135 | 89 | 151 | 3 | 51.46 |
| | PRS | 0 | 35.7 | 23.5 | 39.9 | 0.8 | |
| Major climate events | NRS | 1 | 133 | 112 | 123 | 9 | 50.40 |
| | PRS | 0.3 | 35.2 | 29.6 | 32.5 | 2.4 | |
| Causes of drought and flood happening frequently in present time | NRS | 2 | 170 | 174 | 31 | 1 | 40.67 |
| | PRS | 0.5 | 45.0 | 46.0 | 8.2 | 0.3 | |
| Reasons behind climate change | NRS | 6 | 197 | 141 | 33 | 1 | 38.49 |
| | PRS | 1.6 | 52.1 | 37.3 | 8.7 | 0.3 | |

Note: NC = Not Clear, DK = Do Not Know, NM = No Comment, K = known, WK = Well known.

### 4.3. Cross Tabulation Results

A chi-square test was performed to assess the proportional differences in adaptation status across selected socio-economic variables, farm earning sources variables, and institutional and market accessibility variables. A distribution of gender, level of education, wealth status, experience, and age of farmers is shown in Table 3.

The majority of the farmers surveyed were aged between 31 to 45 years and had never attended a formal educational institution. Of the 378 surveyed farmers, 71.7% reported being illiterate, 11% reported attending primary school, and 10.6% reported only knowing how to write and/or sign their names. The percentages of farmers having some form of higher education, such as having graduated, attended higher secondary, or attended secondary education institutions, were 0.5%, 0.8%, and 6.9%, respectively. Moreover, 48.1% of farmers reported yearly incomes of more than 150,000 taka, 38.1% of farmers reported yearly incomes ranging between 50,000–150,000 taka, while 13.8% reported yearly incomes of less than 50,000 taka. A total of 378 farmers, 349 females & 26 males, were included in the sample. The number of male respondents was significantly smaller in comparison to that of female respondents because a large segment of the male farmer population had left their stations at the moment of the survey to earn secondary income as middle men and rickshaw pullers following the early flash flood. Among all respondents, 69.3% of the females surveyed and 7.1% of the male farmers surveyed had already employed adaptation strategies. The results of the survey indicate that 30.4% of surveyed farmers employed adaptation strategies and owned land, while 46% of the surveyed farmers employed adaptation strategies but did not own land. Moreover, 48.1% of the surveyed farmers earned off-farm earnings, which showed a positive correlation to adaptation, while 75.1% of the surveyed farmers attained credit in the form of loans from different non-governmental organizations (NGOs). In terms of market structure, 62.7% of farmers reported that the current rural market structure worked in favor of their adaptation, as they were able to sell their products in such markets, while 13.8 percent farmers held a negative opinion of the existing rural market structure on adaptation, as they had to go to 'Hat', which is located far from their villages, to sell their products.

In summary, the variables of gender, age, education, family size, income, land ownership, farming experience, off-farm earning, livestock ownership, cultivation of flood-tolerant rice varieties, short duration boro rice, non-rice rabi crops, distance to local market, credit from NGOs, and rural market structure were shown to be significantly correlated to the adoption of different adaptation strategies.

**Table 3.** Cross tabulation of adaptability versus socio-economic variables, farm earning sources variables, and institutional and market accessibility variables.

| Variables' Name | | Cross Tabulation | | | |
| --- | --- | --- | --- | --- | --- |
| | | Positive for Adaptation, *n* (%) | Negative for Adaptation, *n* (%) | Total, *n* (%) | *p*-Value |
| **Socioeconomic Status** | | | | | |
| Gender | Female | 262 (69.3) | 87 (23.0) | 349 (92.3) | 0.028 |
| | Male | 27 (7.1) | 2 (0.5) | 29 (7.7) | |
| Age | | 44.03 (12.757) | 47.27 (13.667) | 44.79 (13.031) | 0.000 |
| Education | Illiterate | 213 (56.3) | 58 (15.3) | 271 (71.7) | 0.001 |
| | Low literacy | 29 (7.7) | 4 (1.1) | 33 (8.7) | |
| | Primary | 22 (5.8) | 21 (5.6) | 43 (11.4) | |
| | Secondary | 21 (5.6) | 5 (1.3) | 26 (6.9) | |
| | Higher secondary | 3 (0.8) | 0 (0.00) | 3 (0.8) | |
| | Graduate | 1(0.3) | 1 (0.3) | 2 (0.5) | |
| Family member | | 5.92 (2.338) | 6.53 (2.629) | 6.06 (2.420) | 0.000 |
| Income (yearly) | Less than 50,000 | 49 (13.0) | 3 (0.8) | 52 (13.8) | 0.000 |
| | Between 50,000–150,000 | 121 (32.0) | 23 (6.1) | 144 (38.1) | |
| | More than 150,000 | 119 (31.5) | 63 (16.7) | 182 (48.1) | |
| Land ownership | Yes | 115 (30.4) | 25 (6.6) | 140 (37.0) | 0.046 |
| | No | 174 (46.0) | 64 (16.9) | 238 (63.0) | |
| **Farm Earning Sources** | | | | | |
| Off-farm earning | Yes | 182 (48.1) | 20 (5.3) | 202 (53.4) | 0.000 |
| | No | 107 (28.3) | 69 (18.3) | 176 (46.6) | |
| Livestock ownership | Yes | 115 (30.4) | 25 (6.6) | 140 (37.0) | 0.046 |
| | No | 174 (46.0) | 64 (16.9) | 238 (63.0) | |
| Cultivation of | | | | | |
| Flood tolerant rice | Yes | 20 (5.3) | 1 (0.3) | 21 (5.6) | 0.037 |
| | No | 269 (71.2) | 88 (23.3) | 357 (94.4) | |
| Short duration boro rice | Yes | 19 (5.0) | 1 (0.3) | 20 (5.3) | 0.045 |
| | No | 270 (71.4) | 88 (23.3) | 358 (94.7) | |
| Non-rice rabi crops | Yes | 6 (1.6) | 6 (1.6) | 12 (3.2) | 0.028 |
| | No | 283 (74.9) | 83 (22.0) | 366 (96.8) | |
| **Institutional Accessibility** | | | | | |
| Credit from NGOs | Yes | 284 (75.1) | 70 (18.5) | 354(93.7) | 0.000 |
| | No | 5 (1.3) | 19 (5.0) | 24 (6.3) | |
| Subsidies for seed after flood | Yes | 23 (6.1) | 2 (0.5) | 25 (6.6) | 0.058 |
| | No | 266 (70.4) | 87 (23.0) | 353 (93.4) | |
| **Market Accessibility** | | | | | |
| Distance to local market (km) | Mean (+SD) | 4.84 (2.087) | 5.48 (1.423) | 4.99 (1.967) | 0.000 |
| Rural market structure | Yes | 237 (62.7) | 47 (12.4) | 284 (75.1) | 0.000 |
| | No | 52 (13.8) | 42 (11.1) | 94 (24.9) | |
| **Total** | | 289 (76.5) | 89 (23.5) | 378 (100) | |

Source: Field survey and author calculation.

## 4.4. Percentage of Farmers Who Have Taken Loans and Employed Adaptation Strategies

Following Figure 2 shows that 69.3% of the surveyed female farmers and 7.1% of the surveyed male farmers had planned for adaptation. Overall, 23% of female farmers and 2% of male farmers demonstrated negative attitudes towards adaptation. A total of 72% of farmers surveyed had acquired loans for adaptation from different NGOs; among them, 66.7% were female. Hence, it can be surmised that in Bangladesh, NGOs offering credit for climate change adaptation are more likely to work with female farmers than male farmers. Gender-based discrimination, particularly violence against women, remains a major concern and an obstacle to real development in Bangladesh. Women suffer from poverty and economic and social disadvantages at much higher rates than men. Women's economic dependency limits their opportunities to protest against disadvantages or to take action against discrimination within their own family and society. As such, many NGOs working in various rural districts have special focus on providing assistance to disadvantaged women [24].

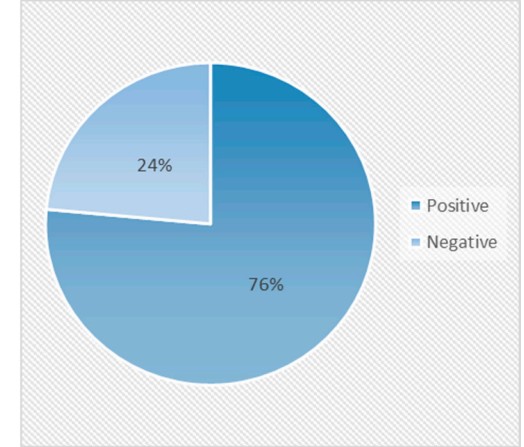

(**a**) Percentage of farmers implementation of adaptation strategies

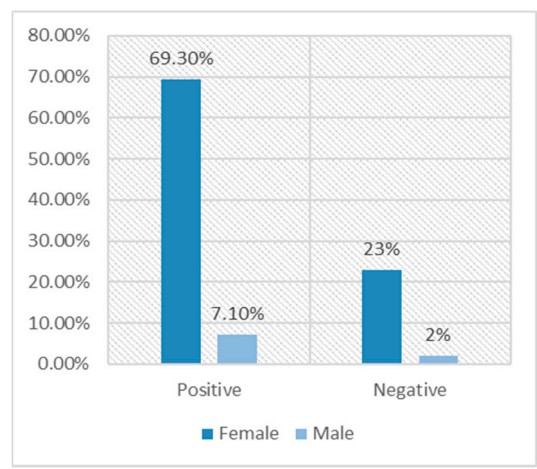

(**b**) Implementation of adaptation strategies by gender

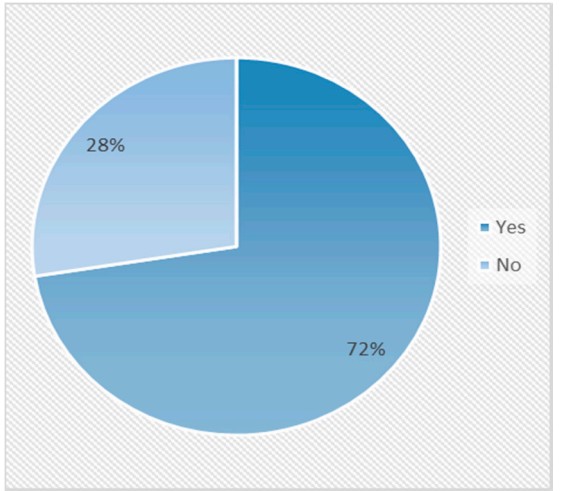

(**c**) Percentage of farmers that have taken credit loans from NGOs

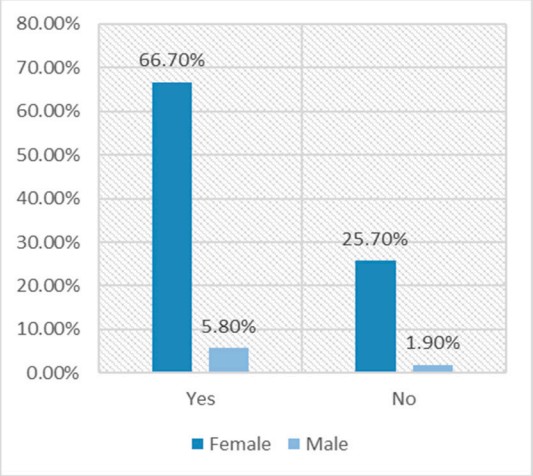

(**d**) Percentage of positive and negative attitudes regarding the acquisition of credit loans

**Figure 2.** Implementation of adaptation strategies by farmers, acquisition of loans to implement adaptation strategies, and attitudes regarding loan acquisition. (**a**) Percentage of farmers that reported implementation of adaptation strategies; (**b**) Implementation of adaptation strategies by gender; (**c**) Percentage of farmers that have taken credit loans from NGOs to implement adaptation strategies; (**d**) Percentage of positive and negative attitudes regarding the acquisition of credit loans to implement adaptation strategies by gender.

### 4.5. Factors Affecting Farmers' Adaptation Practices

Estimated bivariate logistic regression values are given in Table 4. The value of the adjusted $R^2$ indicates that the regression model is able to explain approximately 67.9% of the variation in climate change adaptation strategies.

**Table 4.** Factors affecting farmers' adaptation practices. Here, the dependent variable is the presence of flood-related adaptation practices.

| Variables' Names | Variables' Categories | Adjusted Odds Ratio (AOR) | 95% CI | *p*-Value |
|---|---|---|---|---|
| **Socioeconomic Status** | | | | |
| Gender | Female | 5.498 | 0.925–32.664 | 0.051 |
| | Male | Ref | | |
| Age | | 1.120 | 1.072–1.170 | 0.000 |
| Education | Illiterate | 1.744 | 0.087–5.041 | 0.071 |
| | Know how to write | 0.535 | 0.023–2.585 | 0.069 |
| | Primary | 2.065 | 0.100–4.702 | 0.032 |
| | Secondary | 0.203 | 0.008–2.864 | 0.009 |
| | H. secondary | - | | |
| | Graduate | Ref | | |
| Family size | | 1.033 | 0.907–1.177 | 0.001 |
| Income | Less than 50,000 | 0.122 | 0.033–0.445 | 0.001 |
| | Between 50,000–150,000 | 0.389 | 0.198–0.764 | 0.006 |
| | More than 150,000 | Ref | | |
| Land owner | Yes | 0.868 | 0.830–0.907 | 0.000 |
| | No | Ref | | |
| Farming experience | | 2.542 | 1.304–4.955 | 0.006 |
| **Farm Earning Sources** | | | | |
| Off-farm earning | Yes | 0.195 | 0.110–0.343 | 0.000 |
| | No | Ref | | |
| Livestock ownership | Yes | 0.806 | 0.460–1.414 | 0.053 |
| | No | Ref | | |
| Cultivation of flood-tolerant rice varieties | Yes | 0.117 | 0.002–2.698 | 0.079 |
| | No | Ref | | |
| Cultivation of short duration boro rice | Yes | 1.997 | 0.042–4.448 | 0.025 |
| | No | Ref | | |
| Non-rice rabi crops | Yes | 2.408 | 0.674–5.604 | 0.076 |
| | No | Ref | | |
| **Institutional Accessibility** | | | | |
| Credit from NGOs/GoB | Yes | 18.489 | 6.247–54.725 | 0.000 |
| | No | Ref | - | - |
| Subsidies for seed after flood | Yes | 0.294 | 0.065–1.335 | 0.013 |
| | No | Ref | - | - |
| **Market Accessibility** | | | | |
| Distance to local market (km) | | 1.213 | 1.067–1.379 | 0.003 |
| Rural market structure to sell goods | Yes | 5.175 | 2.930–9.140 | 0.000 |
| | No | Ref | | |

### 4.5.1. Socioeconomic Status

Age group, education, income, and farming experience were found to be significantly correlated with the implementation of adaptation practices. Female farmers were statistically significantly more likely to implement adaptation strategies than male farmers (AOR: 5.498, CI: 0.925–32.664). Likewise, farmers with primary education were shown to be more likely to implement adaptation strategies (AOR: 2.065, CI: 0.100–4.702). It should be noted that the surveyed villages only have education facilities at the primary level. In order to attain an education above the primary level, local people need to go to town or 'Mymensingh' sadar. As such, most local farmers and family members residing in these areas remain uneducated. This lack of schooling translates into insufficient access to information regarding adaptation. The attained results also indicate that there are significant differences in adaptation with respect to land ownership when the socio-economic factor is adjusted. Further, females were shown to

have a 49.8% higher chance of implementing adaptation strategies as compared to male farmers when other socio-economic factors were adjusted.

### 4.5.2. Farm Information

The variable off-farm earning was found to be statistically significant with respect to the adoption of climate change adaptation strategies. Farmers often rely on off-farm income; at the time of the survey, many farmers were working as middle men and rickshaw pullers as a form of earning income after the flash flood. Overall, the ownership of livestock was found to be statistically significant and negatively correlated with adaptation strategies; livestock owners were 20% less likely to adapt in comparison to farmers without livestock. At the time of the survey, most farmers had sold some or all of their livestock due to food shortages. The variable 'cultivation of flood tolerant rice varieties' was found to be statistically insignificant, as no farmers were cultivating flood tolerant rice varieties at the time of the survey. Cultivation of short duration boro rice (AOR: 1.997) was found to be statistically significant, showing a positive correlation with adaptation. The odds ratio of farmers cultivating non-rice rabi crops represents the main effect of adaptation. The value of the odds ratio is 2.408 (95% CI: 0.674–5.604); in short, farmers cultivating non-rice rabi crops were more likely to have adapted as compared to those who had not chosen non-rice rabi crops. In our study, this variable was found to be statistically insignificant as most of the farmers surveyed were unable to plant non-rice rabi crops.

### 4.5.3. Institutional Accessibility and Credit from NGOs

This study showed that the provision of credit by NGOs/GoB was positively correlated with the implementation of adaptation strategies (AOR: 18.489, CI: 6.247–54.725). At the time of the survey, there were many NGOs, including ASA, FIVDB, PKSF, and DADAN, carrying out activities within the surveyed areas. Of these, a large proportion were solely providing credit loans to women stakeholders as part of their gender-focused programming. Micro-credit has become of one of most common and increasingly popular types of NGO initiatives in Bangladesh due to the success this model has had in reaching economically disadvantaged populations, especially women [25]. Although these NGOs were imposing higher interest rates, farmers still chose to attain loans since the funds were highly needed to mitigate the effects of the flood and allow them to adapt after this event. During the flooding crisis, the government (GoB) declared that NGOs would not collect monthly credit instalments from stakeholders for three consecutive months so as to enable farmers to adapt after flood-related losses.

On the other hand, farmers attaining subsidies for seed following the flood demonstrated a 70.6% lower chance of implementing adaptation strategies. However, it should be noted that the majority of the respondents included in the study were not offered subsidies for seed following the flood. Following the flash flood, no actions were immediately taken by the government to provide subsidies for seed, workshops for adaptation planning, or credit to farmers within the studied region, with few exceptions. Although the GoB distributed subsidies for seed to some farmers during the studied period, most of the farmers surveyed could not avail themselves of these opportunities in our study areas. Stronger GoB institutional access in these remote areas could have enhanced the performance of farmers, as well as enabled them to better implement adaptation strategies.

### 4.5.4. Market Accessibility and Rural Market Structure

The ratio of institutional and market accessibility was shown to have a strong positive correlation with the implementation of adaptation strategies. Farmers with access to local market structures located within shorter distances displayed a 21.3% higher chance of implementing adaption strategies (AOR: 1.213, CI: 1.067–1.379). This factor was shown to be statistically significant at a 5% level of significance. The rural market structure in Bangladesh has undergone changes within recent years; until recently, local rural markets used to be the only place where farmers could meet to sell their products. These markets have allowed for the establishment of informal farmer networks, as farmers are able to exchange their knowledge and ideas at rural markets, which are attended daily by farmers

to sell their wares. On the other hand, 'Hats', defined as regional bazars, generally only function once a week. Farmers also attend Hats to sell their products. During the flood, farmers engaged in some adaptation activities to diversify their income, such as fishing, duck rearing, home state gardening, and crop diversification. Within the studied region, farmers were required to commute to the closest Hat in the region, the 'Netrokona bazar', which is located at a fair distance from their local communities, so as to find more customers, as local people within their communities were reluctant or unable to buy products during that time at local rural markets due to the ongoing flood-related financial crisis. The Hat bazaar boasts electricity and solar lighting, which imparts extra advantages for sellers and buyers. Overall, market conditions were found to be the most influential factor for adaptation, as farmers were able to learn and implement adaptation strategies based on their local market and 'Hat'.

*4.6. Farmers' Ranking of Adaptation Practices*

Farmers were asked about their adaptation practices. Common adaptation practices included fishing, home state gardening, and changing crop planting time. Over 40% of the surveyed farmers reported fishing as a secondary source of income, more than 35% stated home state gardening as a practice, and 18% reported they had migrated from farming to non-farming. Less than 5% of surveyed farmers listed duck rearing and working as a middle man as secondary income sources (Table 5).

**Table 5.** Common adaptation practices implemented by farmers by percentage.

| Common Adaptation Practices | Percentage of Farmers |
|---|---|
| Fishing | More than 40% |
| Home State Gardening | More than 35% |
| Migration | 18% |
| Duck Rearing | Less than 5% |

Other adaptation practices implemented by farmers included reducing the number of livestock, implementing a soil conservation scheme, and short-term migration to urban areas. Among all respondents, 373 farmers did not change crop variety, and 369 farmers did not implement soil conservation schemes due to lack of information regarding these practices. Further, 221 farmers switched to livestock production as opposed to crop raising, and 192 farmers sold their livestock in order to adopt to the hostile environment. The results of the WA index showed that reducing the number of livestock ranked first as an adaptation strategy, while short-term migration ranked as 5th (Table 6).

**Table 6.** Farmers' ranking of some adaptation practices.

| Name of Adaptation Practices | Lack of Money (LM) | Lack of Information (LI) | Shortage of Labor (SL) | Lack of Land (LL) | [a] WAI | Rank |
|---|---|---|---|---|---|---|
| Reduce number of livestock | 192 | 2 | 0 | 184 | 1.466 | 1 |
| Change from crop to livestock | 221 | 1 | 0 | 46 | 1.228 | 2 |
| Change crop variety | 2 | 373 | 1 | 2 | 1.008 | 3 |
| Implement soil conservation scheme | 7 | 369 | 1 | 1 | 0.989 | 4 |
| Short time migration to urban area | 327 | 1 | 49 | 1 | 0.269 | 5 |

[a] $WAI = (LM * 0 + LI * 1 + SL * 2 + LL * 3)/N$ Source: Field survey and authors' calculation.

## 5. Discussion

The results of the currently presented cross-sectional study among flood-affected farmers in Bangladesh, which employed a severity index format survey to measure attitudes, perceptions, and knowledge of climate change among this population, corroborate previous research by [26], indicating that most farmers in Bangladesh believe that climate change is a major factor in the adverse conditions currently affecting the Bangladeshi agricultural sector as well as the local climate. The results of this work also corroborate previous studies on rural populations [27–30], substantiating previous findings that marginal farmers from emerging countries comprise one the most vulnerable populations with respect to climate change; in this work, the vast majority of the population under study was found to not have access to sufficient resources to cope with the adverse impact of climate change on their livelihoods.

This study also found that during adverse times, farmers were able to perceive and implement different local adaptation strategies, which helped them to mitigate their losses and supplement their livelihood [2,3,9,31]. Further, in agreement with findings by similar studies in Bangladesh [32,33] and developed countries [26], our study uncovered that farmers were also adopting poor adaptation strategies in times of crisis such as migration—a factor which has further contributed to the low productivity of rice. Our study showed that farmers were unable to sufficiently mitigate the effects of climate change as most farmers found themselves unable to change their crop variety or implement a soil conservation scheme due to poor knowledge and/or resources. These two principal strategies were ranked third and fourth in the preference scale by farmers participating in this study. Moreover, consistent with the other research [9,34,35], in this study, farmers were found to be more likely to adopt local adaptation strategies instead of adopting scientific benchmark strategies such as crop rotation, cultivation of flood tolerant rice, etc. The results of the bivariate logistic regression showed that females were more likely to implement positive adaptation strategies than male farmers; while most male farmers chose to migrate after the flash flood to seek work elsewhere, female farmers preferred to take credit loans from NGOs so as to rebuild or adapt their livelihoods. As such, in agreement with previous findings, this work demonstrated that factors such as gender of the head of household and access to extension services on crop production exert great influence on the type and extent of adaptation strategies undertaken by farmers [9,36–39]. Lastly, the data gathered on off-farm earning strategies demonstrated that farmers supplementing their income through off-farm earning were 19% more likely to sufficiently adapt after adverse times, a finding that is in agreement with previous research by [27]. Rural market structure was also found to be an influential factor for adaptation, which is consistent with other studies in Bangladesh [8]. In agreement with previous calls to action by [40], the findings of this research point at the need for further research on post-disaster impact and mitigation strategies, as merely mitigating the sources of vulnerability that increase the risk of disaster have been proven insufficient to address the needs of this vulnerable population.

## 6. Conclusions

Changing climate remains a major obstacle in the field of agriculture. Changes in climate continue to increase the incidence of early flash floods, a phenomenon which has been increasingly affecting agriculture for many successive years in our study areas. Farmers in Bangladesh have expressed their concern about climate vulnerability, stating that climate change is a major factor impacting the local climate and contributing to the adverse conditions being experienced within the Bangladeshi agricultural sector. This study helped shed light into a wide range of factors that are significantly associated with farmer's adaptation to climate change. Further, our findings indicated that farmers are very interested to work alongside the GoB and NGOs to mitigate the effects of our changing climate by attending workshops and training seminars aimed at learning strategies to reduce climate vulnerability as well as adapt in post-disaster scenarios. The implications of these findings are important since a large number of marginal farmers are struggling to cope with climate change related environmental changes, and to sustain their livelihoods. If corrective actions are not taken, farmers

will continue to face increasingly worse adverse conditions caused by climate change. GoB and NGOs are working for the welfare of Bangladesh, with NGOs (BRAC, ASHA, FIVDB, Suchana Foundation) being pioneers to promote vulnerable communities. GoB organizational wings are also working in this regard along with Non-governmental organizations. Although GoB co-operative wings are present in every Upazila, farmers have only gotten small benefits from these wings. Bangladeshi farmers could benefit from increased access to knowledge through the implementation of more training and knowledge-sharing workshops about climate-adaptive livelihoods through the Bangladesh agricultural office and the Co-operative office, among other governmental offices. Moreover, farmers would benefit from attaining more pertinent climate-related information through radio, mobile short messaging (SMS), and television advertisements, among other forms of communication. Presently, some farmers are receiving mobile SMS; however, this service is limited to some areas. In order to take more advantage of this form of communication, the GoB should extend their service to remote areas as well. Likewise, knowledge-sharing workshops should be extended to vulnerable parts of the country so as to create better awareness among farmers in remote areas. As inhabitants of remote, flood-prone area, farmers have had to struggle for years to secure food, as crops often get destroyed in floods. Access to information and training on techniques to cultivate flood-resilient rice varieties as well as vegetables on elevated land during flood periods could vastly help improve their circumstances. Other factors that may have a positive impact on this population include better implementation of follow-up activities of different projects funded by GoB and NGOs to ensure sustainable development, increased seed subsidies, as well as distribution of better quality seed varieties. Likewise, better coordination among different NGOs and GoB branches would also benefit farmers to this end. Lastly, farmers would also benefit from the implementation of weather- and climate-related technologies, as many existing problems related to agriculture can nowadays be mitigated or prevented through effective use of modern weather technologies. In response to the frequent problems associated with the three consecutive floods to have taken place in these study areas, policy makers should take urgent measures from both inside and outside the embankments, including the following recommended measures: (a) increase the discharge capacity of the rivers; (b) deal with hindrances that do not allow passage of flood waters from inside embankments; (c) undertake careful planning and timely implementation of financial support measures by the GoB or other sources so that farmers will not be heavily indebted; and (d) establish/expand the number of training activities, workshops, and skill development resources within the agricultural sector, particularly in remote areas currently being underserved in this regard. In addition, local institutions such as Union Councils and local community groups should endeavor to work as brokers for social networking through the provision of knowledge, information, and advisory services to enhance adaptation.

**Author Contributions:** K.F.F. had collected, organized and sorted data from flood affected areas and provided results. K.F.F. designed and drafted the manuscript, analysis and results interpretation. M.T.I. and A.A.K. reviewed and edited manuscript to develop the quality of the manuscripts. All authors read and approved the final manuscript.

**Funding:** This research was funded by [SUST Research Center] grant number [PS/2017/17] and The APC was funded by [PS/2018/2/26].

**Acknowledgments:** Authors extend their sincere gratitude to the haor farmers who gave us their valuable time. The first author is grateful to Shahjalal University Science and Technology for funding this project. This research work has been financially supported by the SUST's Research Center Project Fund [PS/2017/17], Bangladesh. We would like to thank the journal editor and three anonymous reviewers for their suggestions and comments. Following their suggestions, we have made enormous improvements in this manuscript. I would like to give special thanks to Luthful Alahi Kawsar and Zobaer Hasan for their help.

**Conflicts of Interest:** There is no conflict of interest among the authors of this manuscript.

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
