# Peer review of "Perceptions, Knowledge and Adaptation about Climate Change: A Study on Farmers of Haor Areas after a Flash Flood in Bangladesh"

_climate, doi:10.3390/cli7070085_

Round 1

Reviewer 1 Report

The paper proposes a severity index to climate change in Bangladesh cities. This is an interesting issue and I think the paper adds a novel contribution in the field. The methodological approach is sound and the selection of variables is well motivated in the literature review section.

I have the following doubts/suggestions.

  1)       Please extend that why this study is important for Bangladesh.  I suggest the authors to properly discuss this issue.

 2)  In the introduction, while the paper is well motivated, I think it would be interesting to explain a bit more in details why and to what extent severity index is in relation to sustainable development of cities.

3)How reliable are those sampling data’s, and did you revise samples or just collect one time?

4) Could you present the how in Bangladesh framers can educate to adopt the climate change strategies in Agriculture.

Author Response

Response to Reviewer 1 Comments

Point 1: Please extend that why this study is important for Bangladesh.  I suggest the authors to properly discuss this issue.

Response 1: I have discussed in conclusion section about why this study is important for Bangladesh.

Point 2: The introduction, while the paper is well motivated, I think it would be interesting to explain a bit more in details why and to what extent severity index is in relation to sustainable development of cities.

Response 2: I have discussed about Severity Index in relation to sustainable development.

Point 3: How reliable are those sampling data’s, and did you revise samples or just collect one time?

Response 3:  Data have been collected during 2017 after devastating flood.

Point 4: Could you present the How Bangladesh farmers can educate to adopt the climate change strategies?

Response 4: I have shown how Bangladesh farmers can educate to adopt the climate change strategies. They can receive subsidies, short massage, training or firsthand lesion from government or Non-government organizations (NGOs).

Reviewer 2 Report

The authors need to provide more detail on Methodology, Results, and Discussion that refer to measuring perception of the farmers and having adaptation planning due to flood.

Author Response

Response to Reviewer 2 Comments

Point 1: More detail on methodology, results and discussion that refer to manuscript perception of the farmers and having adaptation planning due to flood

Response 1: I have added detail methodology, results and discussion in more appropriate manner. Since, Reviewer have given more emphasis on perception and I also showed this, so I think the title should be changed. I would like to give new title “Perceptions, knowledge and adaptation about climate change: A study on farmers of haor areas after flash flood in Bangladesh”.

Reviewer 3 Report

In my opinion the paper is not ready for publication at the current state because the described results don’t provide a significant advance in the knowledge; despite the subject matter is within the scope of the journal, the work conducted is characterized by poor scientific relevance; moreover it is important to point out that the paper is not written in readable English.

Author Response

Response to Reviewer 3 Comments

Point 1: In my opinion the paper is not ready for publication at the current state because the described results don’t provide a significant advance in the knowledge; despite the subject matter is within the scope of the journal, the work conducted is characterized by poor scientific relevance; moreover it is important to point out that the paper is not written in readable English.

Response 1: I have modified my introduction, methodology, results and discussion as well. And I have done English editing also. I have used track changes word documents. I hope reviewer will be able to see all those changes.

Round 2

Reviewer 2 Report

Extensive editing of English language is required, such as "Focuses on adaptation, A few research have..."

Perception is not well defined based on references; for instance, "Adaptation planning are interconnected of Farmers’ knowledge, awareness and perceptions." why?

Author Response

I have edited this manuscript and add my certificate. Editor has changed many sentences that’s why could not add the different color. She used track change review process.

Reviewer 3 Report

The paper is not ready for publication; in particular the methods are not adequately described, the results are not clearly presented and extensive editing of English language and style is required. In fact, there are a lot of unclear and not grammatically correct sentences in the paper; some sentences are listed below:

introduction:

“Bangladesh has agro-zones which are mixed susceptible to scarcity”

“ ……. principally those agro-zones are susceptible to submerge in emerging or evolving countries.”

“The farmers those seem climate change is occurring and is human induced are more likely to perceive adaptation”

“This perception of farmers has been converted to severity index (SI) which is a tool for measuring indexing.”

“In this crisis moment, to mitigate with the existing condition of climate, farmers have taken small scale adaptation and made diversification of production through their locally known knowledge”

“This study, has surveyed  data from the most severely inflated embankment and flood prone district in Bangladesh, aims to provide information on local or native perception to change of climate which principally affect farmers’ adaptation approaches to address proneness of community.”

Section 2.1:

“It damaged nearly ready for harvesting boro rice in about”

“Among the other districts Sumamganj districts was mostly affected according to….”

Section 2.2:

“This study mainly applied cluster sampling in which the census enumeration areas comprising of 100-120 households defined by Bangladesh Bureau of Statistics (BBS) considered as a cluster.”

“The recognize sample size determination formula for the villages was….”

Lines 155-158 are really badly written

Section 2.3:

“……data and information that collected during Nov, year 2017….” The sentence is grammatically wrong !!!!

Section 3.1:

“….Al Hammad and Ashaf (1996) introduced the method of severity index which can be calculated using following formula…..”; the sentence is grammatically wrong   !!!!

Section 4.1:

Lines 230-234 are not grammatically correct

There are other grammatical errors in the paper

Author Response

I have added detail methodology.  

I have edited this manuscript and add my certificate. Editor has changed many sentences that’s why could not add the different color. She used track change review process.

Round 3

Reviewer 2 Report

The authors change the title that refers to perceptions, knowledge and adaptation about climate change, but mention that "Adaptation planning is interconnected to farmers’ knowledge, awareness, and perceptions" (p.4), without any citations. The focus of the study could be further clarified.

English language is minor spell check required such as "Perceptions of climate refers to the way in which climate is regarded..."

Author Response

Point 1. I have added reference of knowledge about climate change, awareness of climate change,and adapting measures which are influenced by socio-economic and others status.

Point 2. I have modified my objectives as well.

Reviewer 3 Report

Accept in present form

Author Response

Point 1: I have checked again whole manuscript and submitted my manuscript through track changes.